# Investigating Ethnic Disparity in Living-Donor Kidney Transplantation in the UK: Patient-Identified Reasons for Non-Donation among Family Members

**DOI:** 10.3390/jcm9113751

**Published:** 2020-11-21

**Authors:** Katie Wong, Amanda Owen-Smith, Fergus Caskey, Stephanie MacNeill, Charles R.V. Tomson, Frank J.M.F. Dor, Yoav Ben-Shlomo, Soumeya Bouacida, Dela Idowu, Pippa Bailey

**Affiliations:** 1Bristol Medical School: Population Health Sciences, University of Bristol, Bristol BS8 2PS, UK; a.owen-smith@bristol.ac.uk (A.O.-S.); fergus.caskey@bristol.ac.uk (F.C.); stephanie.macneill@bristol.ac.uk (S.M.); y.ben-shlomo@bristol.ac.uk (Y.B.-S.); pippa.bailey@bristol.ac.uk (P.B.); 2Southmead Hospital, North Bristol NHS Trust, Bristol BS10 5NB, UK; 3The Newcastle upon Tyne Hospitals NHS Foundation Trust, Newcastle upon Tyne NE7 7DN, UK; ctomson@doctors.org.uk; 4Imperial College Healthcare NHS Trust, London W12 0HS, UK; frank.dor@nhs.net; 5Bristol Health Partners’ Chronic Kidney Disease Health Integration Team, Bristol BS1 2NT, UK; 6Gift of Living Donation (GOLD), London NW10 0NS, UK

**Keywords:** living kidney donation, living-donor kidney transplantation, ethnic disparity

## Abstract

There is ethnic inequity in access to living-donor kidney transplants in the UK. This study asked kidney patients from Black, Asian and minority ethnic groups why members of their family were not able to be living kidney donors. Responses were compared with responses from White individuals. This questionnaire-based mixed-methods study included adults transplanted between 1/4/13–31/3/17 at 14 UK hospitals. Participants were asked to indicate why relatives could not donate, selecting all options applicable from: Age; Health; Weight; Location; Financial/Cost; Job; Blood group; No-one to care for them after donation. A box entitled ‘Other—please give details’ was provided for free-text entries. Multivariable logistic regression was used to analyse the association between the likelihood of selecting each reason for non-donation and the participant’s self-reported ethnicity. Qualitative responses were analysed using inductive thematic analysis. In total, 1240 questionnaires were returned (40% response). There was strong evidence that Black, Asian and minority ethnic group individuals were more likely than White people to indicate that family members lived too far away to donate (adjusted odds ratio (aOR) = 3.25, 95% Confidence Interval (CI) 2.30–4.58), were prevented from donating by financial concerns (aOR = 2.95, 95% CI 2.02–4.29), were unable to take time off work (aOR = 1.88, 95% CI 1.18–3.02), were “not the right blood group” (aOR = 1.65, 95% CI 1.35–2.01), or had no-one to care for them post-donation (aOR = 3.73, 95% CI 2.60–5.35). Four qualitative themes were identified from responses from Black, Asian and minority ethnic group participants: ‘Burden of disease within the family’; ‘Differing religious interpretations’; ‘Geographical concerns’; and ‘A culture of silence’. Patients perceive barriers to living kidney donation in the UK Black, Asian and minority ethnic population. If confirmed, these could be targeted by interventions to redress the observed ethnic inequity.

## 1. Introduction

Living-donor kidney transplantation is the optimal treatment for most people with kidney failure in terms of patient survival, graft survival and quality of life [1,2,3,4,5,6]. The healthcare costs associated with living-donor kidney transplants (LDKTs) are less than for dialysis and deceased-donor kidney transplants (DDKTs) [7,8]. The medium-term risks of donating a kidney are small [9,10,11,12], and the quality of life of donors usually returns to pre-donation levels after donation [13,14]. 

Only 28% of all kidney transplants performed in the UK each year are from a living donor [6], a proportion below that of the USA and the Netherlands [15]. Individuals from Black, Asian and minority ethnic populations in the UK appear to be particularly disadvantaged as they are less likely to receive a LDKT compared to White people with kidney disease [16,17]; only 18% of living donor kidney transplant recipients in the UK between April 2019-March 2020 were from Black, Asian and minority ethnic group backgrounds, despite individuals from these groups constituting 36% of the kidney transplant waiting list [6]. Improving equity in living-donor kidney transplantation has been highlighted as a UK and international research priority by patients and clinicians [18,19]. 

Specific religious and cultural beliefs, as well as a lack of specific knowledge about donation, have been identified as reasons for ethnic disparity in deceased organ donation [20,21]. The barriers specifically encountered by Black, Asian and minority ethnic group patients in accessing LDKTs in the UK are not well described. 

We have previously investigated reasons why individuals who start assessment for kidney donation do not go on to donate in the UK. In this multicentre study, individuals from Black, Asian and minority ethnic groups were more likely to withdraw from donor evaluation [22]. However, transplant candidates and their families often make decisions regarding the suitability of potential donors before they make contact with hospital services. The perceptions of transplant candidates, regarding the suitability of family members for donation, function as an initial stage of donor screening. Transplant candidates are often uncomfortable broaching the subject of organ donation and make assumptions as to why individuals may or may not be able to donate. Transplant candidates may perceive barriers to donation that prevent potential donors from starting donor assessment. It is important to understand these perceptions in order to fully understand barriers to living-donor kidney transplantation. In this multi-centre questionnaire-based study, we investigated the reasons why family members were perceived by kidney patients as unsuitable as living kidney donors, comparing responses between individuals from White and Black, Asian and minority ethnic groups. Ultimately we aimed to identify potentially modifiable barriers to LDKTs specific to the UK Black, Asian and minority ethnic populations that could be targeted to redress the observed disparity.

## 2. Experimental Section

### 2.1. Study Design

We designed this multi-centre questionnaire-based study to investigate the patient-identified and reported reasons potential donors did not donate. We collected both quantitative (checklist item selection) and qualitative (free-text) questionnaire data to gain a greater understanding than that provided by one data type [23]. We collected data on whether participants asked potential donors to donate, whether any offered, and whether any started donor assessment. We collected data from both LDKT and DDKT recipients—LDKT recipients may have had other potential donors who volunteered but did not donate, and we wanted to ensure we captured the reasons for non-donation for all. We compared the responses of Black, Asian and minority ethnic individuals with White individuals to identify barriers that might be specific to Black, Asian and minority ethnic populations and therefore might explain the observed ethnic inequity. 

### 2.2. Participants

The study was based at 14 hospitals in England and Northern Ireland (Appendix A). We obtained from each hospital an anonymised list of all individuals who received kidney transplants between 1/4/13 and 31/3/17, stratified by LDKT/DDKT status. Individuals < 18 years at time of transplantation, or who lacked mental capacity according to the Mental Capacity Act 2005 were excluded. We calculated the study sample size using a variable not analysed here: the patient activation variable [24]. The study was designed to detect a 7-point difference in a continuous measure of patient activation (analysis of this variable not presented here) between LDKT cases and DDKT controls with 90% power, assuming a 5% significance level. The calculation indicated that 170 patients would be needed, and that, therefore, a total of 944 would be needed to allow analyses stratified by Index of Multiple Deprivation rank quintile and allow for 10% missing data. This sample size allows for the detection of a far smaller difference (0.16 Standard Deviation) for a dichotomous exposure or between 6–8% for a categorical outcome [24]. We performed stratified random sampling to select, on average, 110 LDKTs and 110 DDKTs from each site, weighted by the number of transplants performed at each study site. Sex and 5-year age group strata matched sampling was used to try to ensure a similar sample distribution by age and sex.

Between October 2017-November 2018, collaborators at study sites mailed paper questionnaires to participants. Questionnaires were accompanied by an invitation letter, a return postage-paid envelope, and a patient information sheet. A website-address was provided for participants who preferred to complete the questionnaire online. Non-responders were sent a second questionnaire after 4–6 weeks. We extracted anonymised data from returned paper questionnaires at the University of Bristol, and uploaded these onto a secure REDCap database [25].

### 2.3. Questionnaire Content

We have previously reported the development of the questionnaire alongside the findings of a single centre pilot study [24]. Participants were asked to indicate the number of living relatives ≥18 years from a list (spouse/partner, parents, sisters/brothers, children, aunts/uncles, first cousins) as a proxy for their potential living-donor pool. Friends and colleagues were not included, as they contribute very small numbers to the donor pool: between 2006–2017 only 8% of UK living donors were in this category (unpublished data provided by NHS Blood and Transplant to co-author P.B). We asked participants how many relatives had (i) offered to donate, (ii) been asked to donate by the respondent, and (iii) started donor assessment. Participants were asked for the reasons why any of the listed relatives could not donate; individuals were asked to tick all options that applied and were allowed to select multiple reasons from the following list, derived from previous qualitative research into barriers to donation [26]: Age—too old or too young to donate; Health—not healthy enough to donate; Weight—too over or underweight to donate; Location—they live too far away to be able to donate; Financial/Cost—the financial impact of donation would be too much; Job—not able to take the time off work to donate; Blood group—not the right blood group to donate; No-one to care for them after donation. A box entitled “Other—please give details” was provided for free-text entries. Individuals who the respondent considered suitable for donation but who did not donate because another person did were not considered as “not able” to donate. The responses indicated the patient-reported, and therefore the patient-identified, reasons for non-donation.

### 2.4. Main Exposure and Other Demographics

We collected data on self-reported ethnicity, religion, age, sex, and marital status. Participants could select “Would rather not answer” for all demographic questions. Participants indicated their ethnicity according to the UK’s Office for National Statistics (ONS) 2011 census categories [27]: White; Asian/Asian British (Indian, Pakistani, Bangladeshi, Chinese); Black/African/Caribbean/Black British; Mixed/Multiple (White and Black Caribbean, White and Black African, Any other Mixed/Multiple ethnic background); Other (Arab, Any other ethnic group). For the religion variable, participants were asked to select one option from the following: No religion; Christian; Muslim; Jewish; Hindu; Sikh; Buddhist; Other. Age was a categorical variable in 10-year age groups.

### 2.5. Statistical Analysis

We used descriptive statistics to summarise the characteristics of transplant recipients and their reported reasons for non-donation from family members. Black, Asian and minority ethnic group participants comprised “Asian/Asian-British”, “Black/African/Caribbean/Black British”, “Mixed/Multiple ethnic groups”, and “Other Ethnic group”. We derived a binary variable of Black, Asian and minority ethnicity (code = 1) versus White ethnicity (code = 0) as our primary exposure. The Chi2 test was used to compare the characteristics of White and Black, Asian and minority ethnic group participants, and the reasons given for non-donation. We used multivariable logistic regression to describe the association between the reporting of each reason for non-donation with respondent self-reported ethnicity. We used two models: (i) unadjusted and (ii) adjusted for potential confounders. We specified, a priori, potential confounders including sex and age. We considered socioeconomic position as a mediator on the causal pathway between ethnicity and living donation, rather than a potential confounder: we did not adjust for it in our model as this would result in potential over-adjustment and attenuation of the effect of ethnicity. We used robust standard errors to account for clustering within kidney centres. We tested for interactions between ethnicity and age and sex. We identified missing data and described patterns of missingness. We performed both a complete case analysis and a sensitivity analysis using multiple imputation using chained equations to derive 40 imputed datasets per group, for the exposure variable and potential confounders and then combined using Rubin’s rules. All statistical analyses were undertaken using Stata 15 [28].

### 2.6. Qualitative Analysis

Individuals were able to provide free-text qualitative data responses to the question “Thinking about those people you think could not donate a kidney to you, what are the reasons for this?”. All free-text responses from Black, Asian and minority ethnic group respondents were analysed, so no sampling was required. The written free-text responses were typed onto the REDCap database [25]. Free-text responses and participant demographics were then downloaded from REDCap onto an Excel spreadsheet file. NVivo qualitative software was used to facilitate analysis. Data were analysed using inductive thematic analysis [29], as described by Braun and Clarke [30]. After familiarization with the data, sections of text within the responses were coded by assigning descriptive labels. Codes were collated on the basis of shared properties to create initial potential themes, which were then refined. Themes were revisited and finalised during the preparation of the report for publication. Coding and thematic analysis were undertaken independently by both K.W. and P.B. Coding discrepancies were resolved by discussion to enhance rigour and reliability. All themes were reported using a minimum of three illustrative quotes. After completing analysis for Black, Asian and minority group respondents (*n* = 56), a matching number of White participants (*n* = 56) were purposively sampled aiming for diversity in terms of age, sex and socioeconomic status, and qualitative responses analysed for comparison.

The Strengthening The Reporting of OBservational studies in Epidemiology (STROBE) and COnsolidated criteria for REporting Qualitative studies (COREQ) guidelines were used to prepare the manuscript [31,32]. 

### 2.7. Ethical Approval and Consent

We received NHS Research Ethics Committee (REC) (REC reference 17/LO/1602) and Health Research Authority (HRA) approval. A consent form formed the first page of the questionnaire. The study was funded by a Kidney Research UK Project Grant (RP_028_20170302). The clinical and research activities being reported are consistent with the Principles of the Declaration of Istanbul as outlined in the “Declaration of Istanbul on Organ Trafficking and Transplant Tourism”.

## 3. Results

### 3.1. Quantitative Findings

A total of 1240 questionnaires were returned from 3103 patients (40% response). The characteristics of all respondents are described in Table 1. LDKT recipients were more likely to respond than DDKT recipients and women were more likely to respond than men (Appendix A). Study participants appeared to be generally similar to the National population of DDKT and LDKT recipients though the study sample had fewer Black, Asian and minority ethnic group participants (largest difference 9% for DDKT) (Appendix A). Overall, the proportion of missing data was small (< 10% for all demographic variables) (Appendix A).

White participants were older than Black, Asian and minority ethnic group participants, and a greater proportion of White participants were LDKT recipients compared to Black, Asian and minority ethnic group respondents. Black, Asian and minority ethnic group participants were more likely to report having a religion than White participants: of those with a religion, the majority of Black, Asian and minority ethnic group participants reported a religion other than Christianity, whereas the majority of White participants reported being Christian (Appendix A). Black, Asian and minority ethnic group participants reported a larger number of potential donors compared to White respondents (median number of family members ≥ 18 years: 19 versus 16, Wilcoxon rank-sum test *p* = 0.02).

Most participants had not asked any of their relatives to donate (*n* = 848/1181, 71.8%). In total, 81.8% (*n* = 973/1189) reported that one or more relative had offered to donate, with 85.6% of these actually starting donor assessment (representing 14.4% attrition).

Participant responses to the question “Thinking about those people you think could not donate a kidney to you, what are the reasons for this?” differed by ethnicity (Table 2). Black, Asian and minority ethnic group individuals were more likely than White respondents to indicate that family members lived too far away to donate (*p* < 0.001), were prevented from donating by financial concerns (*p* < 0.001), were unable to take time off work (*p* < 0.001), were not the right blood group (*p* = 0.002), or had no-one to care for them after donation (*p* < 0.001). We found no evidence that the proportion of respondents who indicated that age (*p* = 0.96), donor health (*p* = 0.88), or donor weight (*p* = 0.36) were reasons for non-donation differed between White and Black, Asian and minority ethnic group respondents. 

There was strong evidence that even after adjustment for potential confounders of sex and age, Black, Asian and minority ethnic group individuals were more likely than White respondents to indicate that family members lived too far away to donate (adjusted odds ratio (aOR) 3.25 (95% Confidence Interval (CI) 2.30–4.58)), were prevented from donating by financial concerns (aOR 2.95 (95% CI 2.02–4.29)), were unable to take time off work (aOR 1.88 (95% CI 1.18–3.02)), were not the right blood group (aOR 1.65 (95% CI 1.35–2.01)), or had no-one to care for them after donation (aOR 3.73 (95% CI 2.60–5.35)) (Table 3). The associations did not differ substantially between the complete cases analysis and the analyses with missing variables imputed (Appendix A). In total, 11 individuals who had not selected the “Health – not healthy enough to donate” response indicated in the free-text that potential donors had or might develop the same kidney disease as them. In a sensitivity analysis, when these individuals were recoded as selecting “Health” as a reason for non-donation, there was no change in the direction or the size of associations observed in Table 3.

There was a modest suggestion of interaction between sex and ethnicity (likelihood ratio test *p* = 0.03) in the reporting of “no-one to care for them after donation” as a reason for non-donation (Appendix A) so the increased risk seen for Black, Asian and minority ethnic group was only seen in men.

### 3.2. Qualitative Findings

In total, 56 Black, Asian and minority ethnicity individuals provided free-text reasons for potential donor unsuitability: respondent characteristics are presented in Appendix A. Four overall themes were identified (Table 4): (i) Burden of disease within the family, (ii) Differing religious interpretations, (iii) Specific geographical concerns, and (iv) A Culture of Silence. 

#### 3.2.1. Burden of Disease within the Family 

A large number of Black, Asian and minority ethnic group respondents stated that potential donors were unable to donate due to presumed or perceived ill health. Respondents reported a heavy burden of both hereditary and non-hereditary kidney disease precluding donation: 


*“Family history of PKD [polycystic kidney disease]—all siblings, all children and uncles affected.”*
*(Female|50–59 years|Asian|LDKT)*


*“Too old and unhealthy. Heart problem, Diabetes, high blood pressure, inheritance.”*
*(Male|60–69 years|Asian|Sikh|DDKT)*

Participants also reported that health problems were identified during donor assessment that prevented donation:


*“There were genetic issues that were contra-indications such as a cause of cancer which was discovered during screening…”*
*(Male|20–29 years|Asian|Muslim|DDKT)*

#### 3.2.2. Differing Religious Interpretations

Several participants reported that a relative’s religion or faith had prevented them from donating: 


*“Their religion would not allow them to donate a kidney.”*
*(Female|40–49 years|Black|Christian| LDKT)*

However, most participants considered the beliefs as unorthodox, describing what they perceived as a distortion of a religious belief:
“Superstition/religion (distorted beliefs). Myth.”*(Female|60–69 years|Black|LDKT)*
and a discordance between the participants’ and their relatives’ interpretations of their faith:


*“Their religion/faith forbids them to donate… thought they were Christians like me.”*
*(Female|60–69 years|Black|LDKT)*

No participants who reported religion as a barrier to donation for their relatives reported that they shared their relatives’ beliefs. All but one of the respondents who reported religion as a reason for non-donation self-identified as Christian and was Black/African/Caribbean/Black British. 

#### 3.2.3. Geographical Concerns 

Several participants reported relatives being unable to donate due to geographical separation. However, it was not the distance alone that was considered a barrier to donation for some: 


*“While some are abroad they were willing to travel.”*
*(Male|60–69 years|Black|Christian|LDKT)*

Rather, participants reported difficulties with immigration rules:
“Immigration rules can be problematic too.”*(Male|40–49 years|Black|Muslim|LDKT)*
prohibitive financial concerns:
“My blood relatives live outside the UK. The financial cost has been a major issue.”*(Male|50–59 years|Other ethnic group|DDKT)*
and concerns about the quality of post-donation healthcare in their potential donor’s country of residence: 


*“I come from Papua New Guinea and health services are poor. People are afraid of death during and after donating of their kidneys. After operations the care given is not very good and people end up dying. We lost two relatives from sepsis.”*
*(Female|50–59 years| Other ethnic group|Christian|LDKT)*

#### 3.2.4. A culture of Silence 

Several participants described a “culture of silence” around their illness, reporting that their family were not aware they had kidney disease:


*“Are unaware of my current condition.”*
*(Male|20–29 years|Asian|Hindu|LDKT)*

This was reported as a result of some participants personally not disclosing this information to relatives:


*“…my reluctance to show how ill I was, to soldier on, accept my fate and manage accordingly.”*
*(Male|50–59 years|Asian|Sikh|LDKT)*

As well as other family members controlling the disclosure of information to the wider family: 


*“The majority of my extended family do not ‘officially’ know that I am unwell/having dialysis or had a transplant as my parents did not want them to know.”*
*(Male|30–39 years|Other ethnic group|Other religion|DDKT)*

A summary model of barriers identified is presented in Figure 1. 

#### 3.2.5. Responses from White Participants 

Comparing these free-text responses against those from the 56 purposively sampled White participants (Appendix A), only one theme proved common to both White and Black, Asian and minority ethnic group respondents—”Burden of disease within the family”. Two further themes were identified amongst White respondents that were not evident in the Black, Asian and minority ethnic group dataset: (i) Lack of close family relationships—through relationship breakdown or dysfunction and (ii) Protecting others. These themes and illustrative quotes are presented as Appendix A.

## 4. Discussion

The majority of respondents indicated that they had not asked potential donors to donate, suggesting that transplant candidates may make assumptions as to why individuals may or may not be able to donate. Although 81.8% reported that one or more individuals had offered to donate, 14.4% of these participants did not have a potential donor that proceeded to donor assessment. Whilst some of these individuals may have received a DDKT before their potential donor started assessment, others may have been deemed unsuitable for donation by the transplant candidate. These findings highlight the importance of understanding patient-identified reasons as to why individuals are deemed unsuitable as living kidney donors. 

Black, Asian and minority ethnic group participants were more likely than White participants to indicate that family members lived too far away to donate, and to report financial concerns in part linked to geographical distance. The qualitative data provided insight into these identified barriers, and as described they would appear to be surmountable. In the UK, NHS England allows potential donors from overseas to be reimbursed for travel, accommodation and visa costs after the event [33]. However these large “up-front” costs may be prohibitive to potential donors, and previous qualitative research has shown that many patients are unaware of the reimbursement policy [26]. Clarifying UK immigration policy and highlighting the reimbursement scheme may help potential Black, Asian and minority ethnic group recipients access their potential donor pool. 

Previous research by the authors has suggested that Black, Asian and minority ethnic group ethnicity and non-Christian religious affiliation are associated with greater uncertainty in beliefs about living donation [34]. No respondents in our study reported perceiving a specific religion as forbidding living donation. This may reflect the success of work by faith leaders to clarify positions on living donation within the UK, including a new fatwa clarifying Islamic approval of organ donation and transplantation published in the UK in 2019 [35]. However, the participants’ responses indicated that some of their potential donors did perceive religion as a barrier to donation. In particular there were several references to the distortion of religious beliefs being a barrier to donation. This highlights the need to better understand and consider the beliefs of potential donors who belong to non-mainstream religions, who may be outside the remit of denominational faith leaders. 

A “culture of silence” about illness was an important theme identified in responses from Black, Asian and minority ethnic group participants. Although not directly comparable to the UK Black, Asian and minority ethnic group population, qualitative research in African-American LDKT recipients and donors has suggested that restricting disclosure and maintaining privacy of health status can protect against feelings of vulnerability [36], help to maintain self-perception and public identity, and is linked with rejection of the sick role which is sometimes associated with better coping skills in patients with kidney disease [37]. Potential African-American donors have also reported negative responses from family and friends regarding donation, and encouraging the recipient not to disclose their health status may be perceived as a protective act in that context [36]. We found that Black, Asian and minority ethnic group participants have larger potential donor pools than White participants, but this “culture of silence” may mean that Black, Asian and minority ethnic group individuals are less able to access their pool and therefore a LDKT. It may also mean that Black, Asian and minority ethnic group individuals are less able to access their social network during time of chronic illness: lack of social support and lack of an informed social network are associated with reduced access to transplantation [26,38,39,40], and worse transplant outcomes [41]. Interestingly, in White participants, lack of close relationships was identified as reason for non-donation, but this was not reported by Black, Asian and minority ethnic group participants, despite the geographical separation. Strategies to overcome this culture of silence could include interventions that engage with a patient’s social network, such as the Dutch home-education model shown to be effective at increasing access to living-donor kidney transplantation for minority ethnic groups [42]. A focus on the potential benefits to family members from the education session (detection of undiagnosed kidney disease, how to optimise own health) could be emphasised. The use of “live donor champions” may also enable discussions to start: in this approach, a friend or family member is trained to undertake an advocate role, sharing information on the patient’s behalf with the patient’s wider social network [43]. Other approaches that may overcome “cultures of silence” include people with kidney disease, transplant recipients and donors sharing their experiences on an open web-platform such as healthtalk.org (http://healthtalk.org) and the living donation storytelling project (https://explorelivingdonation.org/) [44]. However, such approaches need to be formally evaluated for effectiveness.

Black, Asian and minority ethnic group respondents were more likely to report that potential donors were not the right blood group. Whether this represents true or perceived incompatibility requires further investigation. A single-centre study from the USA in 2002 found that more African-American donors than White donors were prevented from donating due to ABO incompatibility (9.7% vs. 5.6%, *p* < 0.01) [45]; however, to our knowledge this has not been examined in the UK. If found to be true, willingness to participate in the UK Living Kidney Sharing Scheme should be investigated, and participation encouraged.

This was a large, multicentre study utilising both quantitative and qualitative data. The questionnaire was evaluated in cognitive interviews prior to use and then piloted. The proportion of missing data was very small. However, the study has some limitations: (i) There is a risk of self-selection bias given our response rate, although this is comparable to other postal surveys in the UK [46,47] and the 47% response to a survey sent to Dutch and Swedish transplant recipients [48]. There is some evidence that Black, Asian and minority ethnic group individuals may have been under-represented but it is unclear whether the participants in the study would be different in respect to the reported reasons for non-donation. We suspect, if anything they would be more knowledgeable and engaged and so some of our results may underestimate the true associations. We did not have data on the ethnicity of non-responders and so we were unable to ascertain if there was a difference in response rate between the Black, Asian and minority ethnic group and White populations. (ii) Ethnicity can be described as a form of collective identity that draws on notions of ancestry, cultural commonality, geographical origins, and shared physical features. Ethnic identities are social constructs that are fluid across space and time [49]. In this study, ethnicity was coded using the UK’s ONS 2011 census categories, but individuals may self-identify with several or none of the ethnic categories used in government statistics [49]. Any ethnic identify categorisation fails to respect the heterogeneity within a group due to differing cultures, religions, languages, HLA-types, whether a person was born in their place of residence or migrated to it, and for migrants, time resident. We analysed all Black, Asian and minority ethnic group respondents as one group as our sample size prevented analysis by more specific ethnic groups (e.g., Asian-Indian, Black British, Chinese). Study findings should be considered an indicator of a signal that requires further detailed investigation. (iii) The questionnaire was only available in English, as several survey tools had only been validated in English. Findings may therefore not be applicable to patients who do not read English, who may be from White or Black, Asian and minority ethnic group groups. (iv) Participants had all received a kidney transplant and findings may not be generalisable to transplant eligible people active on the transplant waiting list. Qualitative responses were limited to hand-written free-text entries and were all in English, which may have restricted participants for whom this was not a first language. In-depth interviews would allow for further investigation of the issues raised. This study describes patient-reported and patient-identified reasons potential donors were not considered suitable for kidney donation. Patients are gatekeepers to the process, making personal judgements as to suitability: both who to approach and whose offers to accept or decline. However, surveying non-donors about their reasons for non-donation would provide a different and important perspective, although such a study would be ethically and practically challenging [50]. 

## 5. Conclusions

We have identified multiple patient-identified barriers to living kidney donation in the UK Black, Asian and minority ethnic group population, which should be further investigated and addressed to reduce the ethnic inequity in living-donor kidney transplantation in the UK.

## Figures and Tables

**Figure 1 jcm-09-03751-f001:**
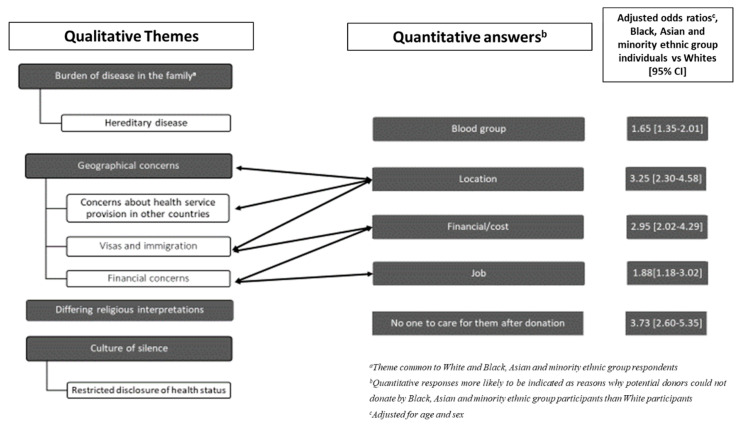
A summary model of barriers to living kidney donation as reported by UK Black, Asian and Minority Ethnic individuals.

**Table 1 jcm-09-03751-t001:** Participant demographics.

Characteristics		Participants (*n* = 1240) *n* (%)
Sex	Female	514 (41.5)
	Male	705 (56.9)
	Missing	21 (1.7)
Type of transplant	Living-donor kidney transplant	672 (54.2)
	Deceased-donor kidney transplant	565 (45.6)
	Missing	3 (0.2)
Age group (years) ^a^	20–29	74 (6.0)
	30–39	137 (11.1)
	40–49	209 (16.9)
	50–59	331 (26.7)
	60–69	299 (24.1)
	70–79	150 (12.1)
	80–89	6 (0.5)
	Missing	34 (2.7)
Self-reported Ethnicity ^b^	White	1027 (82.8)
	Asian/Asian-British	79 (6.4)
	Black/African/Caribbean/Black British	58 (4.7)
	Mixed/Multiple ethnic groups	10 (0.8)
	Other Ethnic groups	24 (1.9)
	Missing	42 (3.4)
Religion	Christian	717 (57.8)
	Hindu	27 (2.2)
	Sikh	13 (1.1)
	Muslim	21 (1.7)
	Jewish	6 (0.5)
	No religion	335 (27.0)
	Other	38 (3.1)
	Missing	74 (6.0)

^a^ No participants aged <20 years. ^b^ UK’s Office for National Statistics 2011 census categories.

**Table 2 jcm-09-03751-t002:** Participant reported reasons relatives could not donate a kidney to them.

Reported Reason Potential Donor not Suitable for Donation	White *n* = 1027, *n* (%)	Black, Asian and Minority Ethnic Group *n* = 171, *n* (%)	White vs. Black, Asian and Minority Ethnic Group Chi^2^ *p*-Value
Age—too old or too young to donate	562 (54.8)	94 (55.0)	0.96
Health—not healthy enough to donate	648 (63.2)	109 (63.7)	0.88
Weight—too over or underweight to donate	152 (14.8)	30 (17.5)	0.36
Location—they live too far away to be able to donate	188 (18.3)	72 (42.1)	<0.001
Financial/cost—the financial impact of donation would be too much	98 (9.6)	40 (23.4)	<0.001
Job—not able to take the time off work to donate	106 (10.3)	29 (17.0)	<0.001
Blood group—not the right blood group to donate	199 (19.4)	51 (29.8)	0.002
No-one to care for them after donation	63 (6.1)	32 (18.7)	<0.001

**Table 3 jcm-09-03751-t003:** Multivariable logistic regression analysis comparing reasons potential donor unsuitability between White and Black, Asian and minority ethnic participants ^a^.

Reported Reason Potential Donor Not Suitable for Donation	Black, Asian and Minority Ethnicities vs. White Unadjusted Odds Ratio (OR) [95% Confidence Interval (CI)]	Black, Asian and Minority Ethnicities vs. White Adjusted for Sex and Age OR [95% CI]
Age—too old or too young	1.00 [0.75–1.34]	0.98 [0.73–1.32]
Health—not healthy enough	1.02 [0.78–1.34]	0.96 [0.71–1.31]
Weight—too over or underweight	1.22 [0.84–1.77]	1.13 [0.78–1.65]
Location—live too far away	3.23 [2.23–4.68]	3.25 [2.30–4.58]
Financial/cost—financial impact of donation would be too much	2.89 [2.07–4.03]	2.95 [2.02–4.29]
Job—not able to take time off work	1.77 [1.15–2.71]	1.88 [1.18–3.02]
Blood group—not the right blood group	1.76 [1.43–2.17]	1.65 [1.35–2.01]
No-one to care for them after donation	3.51 [2.47–4.99]	3.73 [2.60–5.35]

^a^ Complete case analysis.

**Table 4 jcm-09-03751-t004:** UK Black, Asian and minority ethnic participant qualitative analysis themes and illustrative quotes.

Theme	Representative Quote
Burden of disease within family	“Very healthy but slight amount of protein in urine so not able to donate.” (Male, 50–59 years, Asian, Hindu, Living-donor kidney transplant (LDKT) “They all have slight renal problem” (Female, 50–59 years, Black, Deceased-donor kidney transplant (DDKT) “Hereditary illness in the family” (Male, 50–59 years, Asian, DDKT) “Mother and 2 sibling have same condition as mine (1 sister & 1 brother).” (Male, 30–38 years, Black, Christian, DDKT)
Differing religious interpretations	“Their religion/faith forbids them to donate 1. thought they were Christians like me. 2. our culture forbids them to donate… 3. some forbid blood transfusion and the unbelievable reasons for that.” (Female, 60–69 years, Black, LDKT) “Superstition/religion (distorted beliefs). Myth.” (Female, 50–59 years, Black, Christian, DDKT) “Their religion would not allow them to donate a kidney.” (Female, 40–49 years, Black, Christian, LDKT) “Religious/cultural…” (Male, 50–59 years, Asian, Hindu, LDKT)
Geographical concerns	“All of my family apart from my spouse live in Ethiopia and other countries and would not have access to healthcare or the means to come to the UK” (Male, 40–49 years, Black, Muslim, LDKT) “All my people are in Nigeria, some of them, lack of transport to help them home is the problem some of them have.” (Male, 70–79 years, Black, Christian, DDKT) “I had a word with my mum, wife and my son but they couldn’t come to the UK due to financial and other reasons.” (Male, 40–49 years, Black, Christian, DDKT)
A culture of silence	“I did not ask for a donation so do not have a reason.” (Female, 60–69 years, Asian, Sikh, DDKT) “I would not ask my cousins” (Female, 30–39 years, Asian, Muslim, LDKT) “Other 3 cousins from my mother’s half sister do not have PKD but they would not offer, they didn’t before, I would certainly not ask.” (Female, 60–69 years, Other ethnic group, No religion, LDKT) “Are unaware of my current condition.” (Male, 20–29 years, Asian, Hindu, LDKT)

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
