# Peer review of "Investigating Ethnic Disparity in Living-Donor Kidney Transplantation in the UK: Patient-Identified Reasons for Non-Donation among Family Members"

_jcm, 2020, doi:10.3390/jcm9113751_

Round 1

Reviewer 1 Report

This elegant article by Wong and colleagues explores ethnic disparities in living donor kidney transplantation in the UK. The authors nicely examine potential barriers that kidney transplant recipients of Black, Asian and Minority Ethnic Groups (BAME) face when attempting to find a living donor. They used a survey tool given to all kidney transplant recipients at 14 UK hospitals to investigate whether participants had a potential living donor and their perceptions of the barriers experienced by their donors.

Their results are based on 1240 questionnaires, a respectable 40% response rate. Multivariate logistic regression revealed that BAME participants experienced different barriers to living donation compared to White participants. Distance, finances, work responsibility, support groups and ABO typing were barriers that were significantly greater in BAME compared to White participants.

Overall, this is a well written manuscript with sound statistical findings that are relevant to the population studied. There are definite limitations to the study which are nicely explained in the discussion.

  1. It is well known that there is an imbalance in ethnic groups on the wait list compared to those that receive kidney transplantation, especially those from a living donor. Adding wait-list data and comparing that to those that received living and deceased donor kidney transplants would complement this work.
  2. Can the authors clarify if in the recipients of living donor kidney transplants, the data was based on actual living donors or those who did not proceed to donation.

Reviewer 2 Report

Wong and colleagues report the results of a questionnaire in 1,240 patients with kidney failure from 14 UK hospitals to assess barriers to living organ donation. They pay special attention to disparities between “Whites” and a heterogeneous group of patients labeled as “BAME” (Black, Asian, and Minority Ethnic). Their intention is to improve the rate of live donor transplantation in the BAME group by identifying differences in patient-reported barriers to live-donor transplantation between ethnic groups. Overall, they find BAME individuals lived further from potential donors, cited financial concerns, were unable to take time away from work, were not the right blood type, or had no caregivers available post donation than White individuals. I think these are important questions and research to improve access to living donor transplants in ethnic minorities is required, but I am concerned that the manuscript in its current form could reinforce racial stereotypes.

Major concern:

  • Given the delicacy of the topic and the potential for reinforcing racial stereotypes, I think it is essential to include input from patient partners and experts on race, not just in the review of the manuscript, but also its production. If the current authors fill such roles, it should be clearly delineated. The role and viewpoint of these authors should be clearly stated within the methods, results, and discussion for transparency.
  • Is it appropriate to label groups by skin color (ie. “White” vs. “Black”) instead of ethnicity or ancestry (ie. “European-descent” vs. “African-descent”). It should be clearly stated that ethnicity was “self-reported”, even though the process is described. Could participants select multiple ethnicity boxes? Did any? Which box would an individual check if they had a Black mother and a White Father?
  • Is the use of “BAME” supported by scholars who study the effects of race on medical stereotypes? Would evaluations of heterogeneity across BAME subcategories be adequately powered?
  • Instead of comparing the adjusted odds ratio of naming specific barriers between ethnic groups, I think it would be better to describe the effect that each variable has on living donor transplantation rates (a cox survival analysis of time-to-transplantation with those experiencing the barrier compared to those who do not) in ALL patients with kidney failure. You could then ask if the effect size of the barriers change across ethnic groups (ie. is there an interaction between the ethnicity and barriers). Using this strategy would evaluate the barriers, rather than the ethnic groups. The prevalence of each barrier in different self-reported races could be reported, rather than stating the ethnic group is a risk factor for having the barrier.
  • In my opinion, it is inappropriate to label the reasons for non-donation as “patient-perceived”. If the patient perceives it as a barrier, it is a barrier. This is analogous to “the patient endorses chest pain” or the “the patient reports chest pain”. This type of language is pejorative. If the patient has chest pain, the patient has chest pain.
  • Description of sample collection should be improved. A short clear description of the sample size calculation methods would be appreciated. What was the a priori target sample size? What type of power was expected? Was examining ethnic differences part of the original study design? If so, why were 82.8% of questionnaires distributed to White participants?
  • I would encourage the authors to refer to “the living donation storytelling project” (https://explorelivingdonation.org/) which includes as an aim to improve living kidney donation rates in marginalized populations.

Minor comments:

  • Many are pushing for a shift from “renal” to “kidney” and I would suggest considering adopting this change.

Round 2

Reviewer 2 Report

I think the authors have satisfactorily responded to my comments.